# Pro-Environmental Behavior and Climate Change Anxiety, Perception, Hope, and Despair According to Political Orientation

**DOI:** 10.3390/bs13120966

**Published:** 2023-11-23

**Authors:** Ângela Leite, Diana Lopes, Linda Pereira

**Affiliations:** 1Centre for Philosophical and Humanistic Studies (CEFH), Universidade Católica Portuguesa, 4710-362 Braga, Portugal; 2Faculty of Philosophy and Social Sciences, Universidade Católica Portuguesa, 4710-362 Braga, Portugal; s-drmlopes@ucp.pt (D.L.); s-lipereira@ucp.pt (L.P.)

**Keywords:** climate change anxiety, climate change despair, climate change hope, climate change perception, political orientation, pro-environmental behavior

## Abstract

The main objective of this paper is to assess pro-environmental behavior, climate change anxiety, perception, hope, and despair in different political orientations. Our specific aims included to assess the validity of all the instruments used; to assess whether the factor structure of the scales were valid across political orientations; to evaluate their reliability; to assess differences concerning age, gender, and political orientation; to learn the variables that explain pro-environmental behavior; and to evaluate the moderating role of climate change perception, despair, and hope in the relationship between climate change anxiety and pro-environmental behavior. Confirmatory factor analyses (CFAs), multi-group CFAs (to calculate measurement invariance), multiple linear regressions, and moderations were performed. Results showed that pro-environmental behavior and climate change hope achieved the four assessed levels of invariance across different political orientations; climate change anxiety achieved the first three levels of invariance; and climate change perception and climate change despair achieved configural invariance. Climate change anxiety, personal experience with climate change, and climate change perception (total, reality, and consequences) presented higher values for the left political orientation than for the right or the center. Climate change anxiety variables contributed most to explaining pro-environmental behaviors. Hope, despair, and climate change perception (consequences) moderated the relationship between climate change anxiety and pro-environmental behavior. These results open up new avenues for investigation, specifically to understand why high levels of anxiety lead to more pro-environmental behaviors.

## 1. Introduction

### 1.1. Climate Change

Currently, there is a broad consensus that climate change is one of the most serious threats facing humanity [1,2], implying highly marked changes such as abnormal heat and precipitation [3]. Furthermore, climate change has a major impact on animal species, natural resources, the economy, sociopolitical institutions and structures, cultural traditions, and human health [4], undermining the emotional and psychological well-being of those who are directly or indirectly exposed to them [5].

In this way, our planet and our health are intrinsically interconnected, that is, the air we breathe, the food we eat, and the resources we consume for our subsistence are all factors that determine our quality of life and, consequently, our well-being [6]. Impacts recorded in the health area arising from climate change, such as malnutrition, asthma, and chronic respiratory diseases, are increasingly frequent, due in part to the lack of people using alternatives to personal transport such as walking or cycling [7].

According to Palinkas and Wong [8], climate change impacts mental health both directly (for example, heat stroke, and exposure to extreme weather events) and indirectly (for example, economic loss, threats to health and well-being, displacement and forced migration, collective violence and civil conflict, etc.) through three types of climate-related events: acute, subacute, and long-lasting. The primary mental health symptoms accompanying these disasters are post-traumatic stress disorder, depression, and anxiety; and vulnerable populations include women and girls, populations of a lower socioeconomic status, and older people [9].

### 1.2. Pro-Environmental Behavior

The instability of human behavior is considered one of the biggest contributors to environmental problems [10,11]. Examples of this are climate change, loss of biodiversity, and pollution, which cause the deterioration of our ecosystem [10,11,12,13,14]. Therefore, it becomes pertinent to encourage individuals to engage in a variety of pro-environmental behaviors, such as reducing energy consumption, increasing the use of public transport, and changing agricultural practices, which are seen as the preponderant components of maximizing positive outcomes and reducing the severity of environmental threats [10]. Thus, the ecological balance of the Earth strongly depends on the pro-environmental behaviors of human beings, and it is necessary to understand the behaviors that mitigate and/or aggravate this problem [11].

The often accepted definition of pro-environmental behavior is intentional action that can reduce the negative impact on the environment [14,15], being a concept of great interest to psychologists [16]. This includes different types of behavior such as recycling, using public transport, waste management, and energy consumption. This environmental protection can be promoted indirectly through environmental citizenship behaviors and/or policy support; in this way, individuals can protect the environment through their own individual efforts, such as time and energy management [15].

### 1.3. Climate Change Anxiety

Climate anxiety and eco-anxiety are gaining prominence around the world as people become increasingly aware of the current and future global threats associated with our planet [17,18,19,20]. A complete understanding of climate anxiety must encompass relational, psychosocial, cultural, ethical, legal, and political factors [18] and can be related to a variety of emotions, including worry [21], fear [22], anger [23], sadness, despair, guilt and shame [24], or hope [25].

Clayton [26] noted that weather-related anxiety can be an appropriate reaction to a realistic threat, or it can be excessive and disproportionate; appropriate levels of anxiety can serve as motivators, providing impetus for individuals to take action to address climate threats. However, as also observed by Hrabok et al. [27], some people may fall into helplessness and despair if they believe that their personal or collective acts are insufficient to deal with the challenge of climate change. Climate anxiety can become clinically significant when it is uncontrollable and begins to interfere with the individual’s ability to sleep, work, or socialize [26]; also, this anxiety can become excessive, i.e., highly disproportionate to the actual level of threat [27]. Participants in the Ojala study [28] described their worries about climate change consistently with symptoms of clinical anxiety, including feelings of being overwhelmed [29], panic [30], rumination about negative emotions of guilt, and worry [31].

### 1.4. Climate Change Perception

The question that can be asked is: Why are people all over the world not mobilizing en masse to stop climate change? A possible explanation is related to the perceptions of individuals, that is, climate change not being considered dangerous enough to create a global public response [32]. Individual perceptions of climate change have been widely studied over the last three decades, with the literature trying to understand how different individuals’ perceptions of the climate have contributed to how they perceive and/or get involved in tackling climate change [33]. While most people agree that climate change is real [34], they may also have different beliefs about the extent to which climate change is caused by humans and what its future consequences will be, as well as where and when these consequences will happen [35]. These same perceptions about climate change play a major role in the support that individuals give to climate policies and their involvement in mitigating and/or adapting to them [35].

Climate change can be perceived as negative, neutral, or even positive [35]. However, its impacts are predominantly negative [36], as are most perceptions of it [37]. Perceptions considered negative in relation to climate change are, for example, increased anxiety, and feelings of threat or anguish, among others [38].

### 1.5. Climate Change Hope

Although there is no consensus on how to define hope, Lazarus [39] explains that hope is an emotion related to a pattern of cognitive evaluation, in which it is expected that something desired will come true, even if the probabilities are not high, translating into a way of dealing with negative circumstances in order to find a solution.

In the study by Ojala [28], hope in relation to climate change was observed in three different ways: through strategies focused on meaning, in which individuals trusted different stakeholders, such as politicians or researchers; through a positive reassessment, in which the problem was recognized and participants had the opportunity to change perspectives and examine positive trends in relation to climate change; and through problem-focused confrontation, in which individuals had faith in their own ability and the ability of others to do something about climate change [28]. In addition, Ojala [28] defined two dimensions of hope: “constructive hope” and “hope based on denying the severity of climate change”. Constructive hope was associated with confidence that climate change can be mitigated with collective action, thus having a positive relationship with pro-environmental behavior, political support, and political commitment [40], while the second dimension of hope (denial of climate change’s gravity) was positively related to doubts about climate change and negatively related to pro-environmental actions [40,41]. Therefore, hope can be positively and negatively related to climate change, depending on its characteristics.

### 1.6. Climate Change Despair

The word despair derives from the Latin term “desperare”, which means “descend from hope”. Despair can affect social contexts, including social support networks and communities [42], which can trigger emotional, cognitive, behavioral, and even biological changes, such as increasing the likelihood of diseases that may result in death [43].

First of all, it must be understood that our emotions and feelings are the engines of despair, which may include sadness related to things that have never been seen before by human beings, such as species that became extinct before our time [44]. Despair can also be related to things that we know or fear we will one day lose, for example, species and/or habitats with which we have a lot of interactions, and also things that we have to give up in our lives, such as energy based on fossil fuels and single-use plastics [44].

In addition, there are often feelings of impotence in the face of large-scale adversities, which end up culminating in despair regarding what we should do in the present and what will happen in the future; this variety of feelings affects us emotionally and ends up harming our well-being [44,45]. Moreover, despair in the face of an environmental crisis often leads to coping mechanisms and maladaptive factors, such as avoidance and denial [46], making it necessary to filter out a lot of information that comes to us as it becomes difficult to manage our emotions when reading article upon article about the broadest environmental losses and government failures [44].

### 1.7. Political Orientation

Sociopolitical factors have proven to be an element of great importance, since they are responsible for the convictions of climate change based on social norms, as well as cultural, religious, and moral values [47]. High levels of skepticism about climate change are largely due to an active and well-funded disinformation campaign (mainly by conservatives and fossil fuel-based industries) aimed at criticizing the scientific understanding of the climate problem and discrediting scientists [48]. Thus, this skepticism towards climate change has also become a political issue, increasing polarization [49].

According to the literature, political conservatives tend to dismiss the catastrophic potential of climate change, refusing to attribute its cause to human activities, which undermines the much-needed mitigation effort [33,48,50]. Poortinga et al. [33] found that conservatives tend to value government regulations less, while liberals are more likely to tolerate or even welcome a greater government role in promoting public welfare [51]; this is corroborated in the studies of Hornsey et al. [52], Wang and Kim [53], and Whitmarsh [54]. Consistently, research over the past two decades has reported growing associations between left–right ideology and the environment [55,56,57,58].

### 1.8. Relationship between Pro-Environmental Behaviors and Climate Change Anxiety, Perception, Hope, and Despair

The anxiety surrounding climate change may be linked to constructive climate change problem solving, by making individuals alert to the danger and motivated for action, inspiring them to find possible solutions to anxiogenic situations, and serving as a source of motivation to stimulate their environmental commitment [59,60]. In an investigation from the American Psychological Association [61], people who reported experiencing eco-anxiety were twice as likely to say they are motivated to change their behavior in order to reduce their contribution to climate change compared to those who did not. In contrast, Clayton and Karazsia [17] concluded that climate anxiety was neither positively nor negatively correlated with behavior.

With regard to the perception of climate change, it is known that the more people believe that climate change is happening, the more they are willing to act [62]. A major obstacle to motivating climate change action is the fact that, for many people, the phenomenon seems impersonal and distant in space and time. In the study by Li et al. [63], respondents who stated that the current day was hotter than normal believed more strongly in global warming, expressed greater concern, and exhibited more pro-environmental behaviors, such as donating more money to institutions related to climate support.

Concerning hope in relation to climate change, constructive hope [28] was associated with confidence that climate change can be mitigated via collective action, thus having a positive relationship with pro-environmental behavior, such as supporting climate policies and political commitment [40]. With regard to despair in relation to climate change, feelings of impotence often arise in the face of large-scale adversities, which culminates in despair surrounding what we should do in the present and what will happen in the future [44,45]. Desperation in the face of an environmental crisis often leads to coping mechanisms with maladaptive factors such as avoidance and denial [46], causing a barrier to the adoption of pro-environmental behaviors [44].

Stevenson and Peterson [64] consider that both concepts (despair and hope) have a future-oriented nature; one (hope) increasing and the other (despair) decreasing pro-environmental behaviors. As Stevenson and Peterson [64], we well as McKinnon [65] and Nairn [66], consider, while hope usually motivates individuals to join sustainability-promoting groups, exhaustion experienced as hopelessness and despair can be a demotivating factor, leading to the possible worsening of climate change [65,66]. Stevenson and Peterson [64] warn future generations of climate activists to avoid the counterproductive effects of despair. Concern for future generations with regard to climate change is the origin of the creation of the eco-generativity concept [67]. “Eco-generativity refers to the capacity of individuals to contribute to the preservation of the environment and promote sustainable practices for the benefit of future generations” [67] (p. 2). One way to support future generations in dealing with climate change is precisely to teach them how to manage despair and harness hope.

### 1.9. Relationship between Pro-Environmental Behavior and Sociodemographic Variables

There are some other factors that influence pro-environmental behavior, such as sociodemographic factors, institutional factors, economic factors, social and cultural factors, motivation, environmental knowledge, awareness, values, attitudes, emotion, responsibility, and priorities [15]. Initially, analyses of differences in pro-environmental behavior focused on sociodemographic factors, such as sex, age, education, marital status, place of residence, and personal economic situation; in this way, women exhibit a tendency towards higher levels of cooperation and compassion, predominantly influenced by their roles as caregivers. This inclination results in an increased awareness of environmental issues and, consequently, in a higher likelihood of engaging in pro-environmental behaviors [15].

On the other hand, individuals with higher levels of education and younger ages tend to be more concerned about the environment, being more aware of the possible damage caused by climate change and more familiar with pro-environmental behavior; the same case can be seen in the married population, which tends to pay more attention to environmental problems compared to the single population, since married people are more concerned with the future environmental conditions of the next generation; the same applies to individuals with a higher socioeconomic level [15].

### 1.10. Relationship between Pro-Environmental Behavior and Politic Orientation

Ideology is a system of interconnected values that each play key roles in the personal and social life of individuals, influencing their attitudes and actions over time [50]. However, this can also prevent the updating of their thoughts and actions, which may be the basis for the growing skepticism towards climate change despite excessive scientific evidence [50].

Political ideology greatly influences pro-environmental behavior [68]. The views that individuals construct of the world, as well as their ideologies, generally solidify when individuals are in their twenties [69], guiding their perceptions and behaviors in various contexts. Climate change has become an increasingly politicized and polarized issue, with conservatives (right-wing) being more skeptical of climate change and less willing to act against it than liberals (left-wing) [68].

In fact, political ideology was found to be the most robust predictor of climate change skepticism; “the saliency of political conservativism is reflective of the influence of a broad right-wing disinformation campaign that has been pushing climate change denial” [70] (p. 3). In Western Europe, there is a linkage between individuals’ left–right self-placement and their climate attitudes, and European non-voters are less worried about climate change than voters [71].

### 1.11. Gap, Objectives, and Hypotheses

There are no instruments that assess psychological aspects related to climate change in the Portuguese population. As climate change is an unavoidable issue, in relation to which we have to take a stance, this lack of valid and reliable instruments is a gap that must be filled. Furthermore, as climate change is an issue whose approach requires political intervention, it is important to understand how political ideology interferes with the adoption of pro-environmental behaviors. The concepts of interest are: pro-environmental behavior (actions and activities that individuals undertake in order to contribute positively to environmental sustainability and protection) [64], climate change anxiety (emotional and psychological distress that individuals experience due to concerns about climate change and its potential impacts on the planet) [17], climate change perception (people’s awareness, beliefs, attitudes, and emotions related to the phenomenon of climate change) [35], climate change hope (the belief that collective efforts can make a positive difference and lead to a more sustainable and resilient future) [64] and climate change despair (feelings of hopelessness, anxiety, and distress that individuals experience when contemplating the severity and consequences of climate change) [64], across different political orientations.

The main objective of this paper is to assess pro-environmental behavior, climate change anxiety, perception, hope, and despair across different political orientations. To achieve this aim, several specific aims were established: (a) to validate all the scales for the Portuguese population; (b) to assess whether the factor structure of the scales were valid across different political orientations through measurement invariance; (c) to evaluate their reliability; (d) to assess differences concerning age, gender, and political orientation; (e) to understand which variables explain each item of the pro-environmental behavior subscale; and (f) to evaluate the moderating role of climate change perception, despair, and hope in the relationship between climate change anxiety and pro-environmental behavior. The hypotheses of this study are: (1) the study is expected to find good model adjustment of the instruments to the Portuguese population [17,35,64]; (2) the instruments are expected to be invariant in relation to political orientations [68,69,70,71]; (3) the study is expected to demonstrate the reliability and validity of the instruments used [17,35,64]; (4) the study is expected to find differences in the instruments’ mean values depending on sociodemographic and political variables [15,72,73,74,75,76,77,78]; (5) the variables of climate change anxiety, perception, hope, and despair are expected to contribute to explaining pro-environmental behaviors [17,28,44,45,46]; (6) the variables changes in perception, despair, and hope are expected to play a moderating role in the relationship between climate change anxiety and pro-environmental behavior [59,60,61,62,63,64,65,66,67,68].

## 2. Materials and Methods

### 2.1. Procedures

This study was submitted to the Scientific Council of the Portuguese Catholic University, having been approved. In order to carry out this study, authorization was requested from the original authors of the respective scales to be validated for the Portuguese population. Then, the instruments to be validated were translated and back-translated according to the International Test Commission (ITC) guidelines for translating and adapting tests [79] and the back-translation procedure [80]. One of the scales, the PEBS, was slightly modified so that the study could be extended to other age groups. A questionnaire was also applied to ten participants, with the aim of detecting possible difficulties regarding their understanding of the items, which did not happen. Data collection took place between February and March 2023.

An informed consent form was given to all participants, which included the study’s objective, as well as an explanation of the investigation procedure and confidentiality and anonymity issues. Participants volunteered for this study and could have withdrawn at any time. Taking into account that the instruments were applied in an online context, a data collection platform was used, Google Forms, having chosen the most non-probabilistic sampling method via snowball sampling through social networks.

### 2.2. Measures

#### 2.2.1. Sociodemographic Questionnaire and Political Questions

A sociodemographic questionnaire aimed to collect information about the sample, introducing variables such as age, gender (1—female; 0—male), marital status (1—single; 2—married/de facto partner; 3—divorced/separated), relationship status (0—not in a romantic relationship; 1—in a romantic relationship), whether or not they have children (1—yes; 0—no), level of education (1—less than 4th year; 2—4th year; 3—6th year; 4—9th year; 5—12th year; 6—higher education), professional status (0—professionally inactive; 1—professionally active) and, finally, political ideology (1—extreme left; 2—left; 3—center; 4—right; 5—extreme right).

#### 2.2.2. Pro-Environmental Behavior Scale (PEBS)

There are several scales that assess pro-environmental behaviors; the choice of this instrument was mainly due to the fact that, on the one hand, it is brief and, on the other hand, it allows the assessment of different dimensions (domestic behavior, information-seeking behavior, and transportation choice). The PEBS was developed by Stevenson and Peterson [64] with the aim of measuring and evaluating pro-environmental behaviors in the face of climate change. This scale consists of ten items, and is answered using a Likert scale of 5 points, where 1 corresponds to “never” and 5 corresponds to “always”. A higher score translates to the adoption of more pro-environmental behaviors. The scale contains three subscales: “household behavior” (items 1, 2, 3, 4, 5); “information-seeking behavior” (items 6, 7, 8); and, “transportation choice” (items 9 and 10), whose internal consistencies are α = 0.69, α = 0.66, and α = 0.64, respectively. This scale has a global Cronbach’s alpha of 0.74. The “household behavior” subscale tries to understand how pro-environmental behavior is exhibited at home (for example, “Turn off the lights at home when they are not in use”). The second factor, “information-seeking behavior”, concerns items such as, for example, “Talk with my parents about how to do something about environmental problems”. Finally, the third subscale, called “transportation choice”, corresponds to two items related to the type of transport used (for example, “Walk for transportation”) [64].

#### 2.2.3. Climate Change Anxiety Scale (CCAS)

The CCAS assesses the anxiety generated by climate change. This scale was developed by Clayton and Karazsia [17] and consists of 22 items, in which participants respond using a Likert scale of 5 points, where 1 corresponds to “never” and 5 corresponds to “almost always”. In this instrument, a high score means greater anxiety about climate change. The scale contains four subscales: (1) “cognitive and emotional impairment”; (2) “behavioral commitment”; (3) “personal experience”; and (4) “functional impairment”, whose internal consistencies, ranging from good to excellent, are α = 0.97, α = 0.79, α = 0.86, and α = 0.94, respectively. This scale comprises an overall Cronbach’s alpha of 0.87. Subscale 1 of this scale (includes 8 items: items 1 to 8) concerns cognitive and emotional impairment in response to climate change, reflected in ruminations, difficulties sleeping or concentrating, nightmares, or crying (for example, “I have nightmares about climate change”). Subscale 2 (includes 6 items: items 9 to 14) refers to behavioral commitment: not just a commitment to sustainable behavior, but an understanding of the meaning of their behavioral response (e.g., “I recycle.”). Subscale 3 (includes 3 items: items 15 to 17) corresponds to personal experience regarding climate change (for example, “I have been directly affected by climate change.”). Subscale 4 (includes 5 items: items 18 to 22) addresses functional impairment; high values in this subscale indicate that concern about climate change interferes with the person’s ability to work or socialize (for example, “My friends say I think too much about climate change.”) [17]. Although the personal experience subscale and the behavioral commitment subscale do not address the topic of anxiety, but rather behaviors related to climate change, they contribute to a global dimension that assesses anxiety in the face of climate change.

#### 2.2.4. Climate Change Perception Scale (CCPS)

The CCPS was developed by Valkengoed and collaborators [35] with the aim of measuring individual perceptions of climate change. This scale consists of eight items, which are answered using a Likert scale of 7 points, where 1 corresponds to “strongly disagree” and 7 corresponds to “strongly agree”. Among these values, the individual chooses the one that best describes his stance. A high score means a greater perception of the fact that climate change is truly real, caused by humans, and has negative consequences. The scale contains three subscales: “reality” (items 1, 2); “causes” (items 3, 4, 5); and “general consequences” (items 6, 7, 8), whose internal consistencies, ranging from good to excellent, are α = 0.79, α = 0.91, and α = 0.86, respectively. The “reality” subscale allows us to understand the extent to which individuals believe that climate change is, in fact, happening (for example, “I believe that climate change is real”); this factor contains two of the eight items present in the scale. The second factor, “causes”, assesses the extent to which people point to human causes versus natural causes to explain climate change (e.g., “Climate change is mostly caused by human activity”), and includes three items. Finally, the third factor, named “general consequences”, consists of three items, and aims to understand the extent to which people perceive the consequences of climate change as negative or positive (for example, “The consequences of climate change will be very serious”) [35].

#### 2.2.5. Climate Change Hope Scale (CCHS)

The CCHS was developed by Stevenson and Peterson [64] with the aim of understanding how participants build and maintain hope in relation to climate change (e.g., “If everyone works together, we can solve the problems caused by climate change”). This is a brief unidimensional scale composed of 8 items, each of which is answered using a Likert scale of 7 points, ranging from 1 corresponding to “totally disagree” and 7 corresponding to “agree”, with an additional choice of “I do not see climate change as a problem”; participants who choose the latter option will be excluded from the study. There are no inverted items. A high score translates to a greater hope for climate change. This instrument revealed to have an acceptable internal consistency, with a Cronbach’s alpha of 0.75 [64].

#### 2.2.6. Climate Change Despair Scale (CCDS)

The CCDS was developed by Stevenson and Peterson [64] with the aim of measuring despair in the face of climate change (e.g., “Climate change is such a complex problem that we will never be able to solve it”). This one-dimensional scale consists of four items, to which participants respond using a Likert scale of 7 points, where 1 corresponds to “strongly disagree” and 7 corresponds to “agree”, with an additional choice 0 of “I do not see climate change as a problem”; subjects who choose this latter option will be excluded from the analysis. A high score translates into a higher rate of despair concerning climate change. In addition, the Cronbach’s alpha result for this scale turned out to be lower than ideal (α = 0.59), however, it was considered acceptable by the authors for exploratory studies [64].

### 2.3. Data Analysis

Our preliminary analyses included a descriptive analysis of sociodemographic characteristics and the questionnaire concerning political orientation. Skewness (SI < 3) and kurtosis (KI < 10) values assured the normality of the items’ scales [81].

Confirmatory factor analyses (CFAs) were performed to assess the model fit of the instruments used in this study for the Portuguese sample. The indicators of the goodness of fit were: root mean square error of approximation (RMSEA), the comparative and incremental fit indices (CFI and IFI, respectively), and the standardized root mean square residual (SRMR). If the CFI and the IFI were ≥0.95, the RMSEA ≤ 0.05, and the SRMR ≤ 0.05 [82], an excellent fit was achieved. If the CFI and the IFI were ≥0.90, the RMSEA ≤ 0.08, and the SRMR ≤ 0.10, an acceptable model was achieved. Other indicators were: Satorra–Bentler chi-square (χ^2^), general model significance (*p*), and relative chi-square (χ^2^/DF) (very sensitive to sample size) [83].

Multi-group CFAs were conducted to evaluate whether the factor structure of the scales remained valid across political orientations; four levels of measurement invariance were analyzed: configural (highest item score is in the same factor across groups); metric (item factorial scores are equal across groups); scalar (item intercepts are equal across groups); and error variance invariance (item measurement invariance is equal across groups). The proceeding step-by-step constrained models were assessed through the differences between pairs of nested models (Δ) in their RMSEAs, CFIs, and SRMRs. A change ≥0.01 in the CFI, ≥0.015 in the RMSEA, and ≥0.03 in the SRMR confirmed a deterioration in the model fit when evaluating for measurement invariance [84].

Pearson (continuous variables) and Spearman (ordinal and nominal) correlations were calculated. Correlations between 0 and 0.3 are weak, between 0.3 and 0.5 are moderate, between 0.5 and 0.7 are strong, and between 0.7 and 1 are very strong, either positive or negative [85]. To evaluate the model reliability, convergence, and discriminant validity, Cronbach’s alpha coefficients, composite reliability (CR 0.70 or higher indicates good model reliability), average variance extracted (AVE 0.50 or higher means proper convergence), and square root of the AVE (higher than the highest correlation with any other latent variable) were used [86].

To compare means, the independent means t-test (two groups), and F-test (more than two groups) were used. Cohen’s d and eta squared effect-size were also used accordingly in the tests used [87]. Cohen guidelines [87] were used to interpret the results.

To determine the variables that predicted the pro-environmental subscales, three multiple linear regressions were carried out. Also, simple moderations were performed to assess the moderating roles of climate change perception, hope, and despair in the relationship between climate change anxiety and pro-environmental behavior. The statistical significance level was set at 0.05. Statistical analysis was performed using SPSS version 28, PROCESS, and AMOS version 28.

## 3. Results

### 3.1. Sample

The sample consisted of 535 participants, mostly female (*N* = 402; 75.1%), with a mean age of 25.73 years (*SD* = 10.83; minimum 18 years and maximum 87 years). A total of 312 (58.3%) participants did not have a university education, and the rest did. The sample was mostly made up of single people (*N* = 457; 85.4%), without children (*N* = 456; 85.2%), who were professionally active (students and workers) (*N* = 495; 92.5%). When asked about their political orientation, the sample proved to be quite balanced, with 163 people on the left (30.5%); 183 on the center (34.2%); and 189 on the right (35.3%).

### 3.2. Descriptive

Appendix A presents a description of the instruments used in this study (pro-environmental behavior scale; climate change anxiety scale; climate change perception scale; climate change hope scale; and climate change despair scale). The results of some items of the climate change perception scale are slightly above the reference values, although the majority of the items showed a normal distribution. For each scale, the items with the highest and the lowest values were, respectively: in the pro-environmental behavior scale, items 1 and 10; concerning the climate change anxiety scale, items 11 and 4; regarding the climate change perception scale, items 8 and 3; in the climate change hope scale, items 6 and 7; and for the climate change despair scale, items 2 and 4 (Appendix A
Table A1). Concerning their reliability, all scales presented Cronbach’s alpha values and Omega values above the corresponding reference ones.

### 3.3. Validation

The aim of this section is (a) to validate all the scales for the Portuguese population; (b) to assess whether the factor structure of the scales were valid across different political orientations via measurement invariance; (c) to evaluate their reliability; and (d) to assess differences concerning age, gender, and political orientation.

#### 3.3.1. Pro-Environmental Behavior Scale

(a)A confirmatory factor analysis of the original model proposed by the authors was carried out. In Table 1, the model results are presented, being that the model with three factors, ten items, and three correlations between errors (theoretically supported) is the one that presented the best fit.

(b)Results from the measurement invariance of the PEBS across different political orientations are displayed in Table 2. Configural invariance according to political orientation was confirmed during the first step of the multi-group CFA. The small changes in the fit indices in the next steps also supported metric invariance according to political orientation. In addition, the increase in the level of measurement invariance at the subsequent steps did not present a significant deterioration of the models’ fit; also, error invariance across political orientation was achieved, providing strong evidence that the PEBS operates similarly across different political orientations (left, center, and right). Most of the differences were below 0.01, supporting different levels of measurement equivalence between political orientations (Table 2).

(c)Reliability indices results for the PEBS’ factors are displayed in Table 3. No differences between the Cronbach’s alpha (α) and McDonald’s omega (ω) were observed, except for transportation choice, whose McDonald’s omega could not be calculated because this factor only contains two items. This factor also presents a very low value of Cronbach’s alpha. In spite of this, the PEBS is a reliable measure. In addition, the composite reliability, average variance extracted (AVE), square root of AVE, mean, and standard deviation were calculated (Table 3), and almost all of these values were within the reference range.

(d)Statistically significant differences were found in relation to gender with regard to the household behavior (*t*(198,423) = −2.234; *p* = 0.027; *d* = 0.662; men, *M* = 4.06, *SD* = 0.75; women, *M* = 4.22, *SD* = 0.63) and information seeking (*t*(533) = −3.512; *p* < 0.001; *d* = 1.003; men, *M* = 2.42, *SD* = 1.01; women, *M* = 2.77, *SD* = 1.00) subscales; women presented higher values than men. Age correlated positively and significantly with the household behavior (*r* = 0.093; *p* = 0.032) and information seeking (*r* = 0.154; *p* < 0.001) subscales.

#### 3.3.2. Climate Change Anxiety Scale

(a)In order to validate the climate change anxiety scale for this population, a confirmatory factor analysis of the original model proposed by the authors was carried out. The models found are presented in Table 4, and the one with the best fit was the four-factor model with 22 items, which had six correlations between errors of the same factor and, therefore, is theoretically supported.

(b)Results from the measurement invariance of the CCAS across different political orientations are displayed in Table 5. Configural invariance according to political orientation was confirmed during the first step of the multi-group CFA. Small changes in the fit indices in the next steps also supported metric invariance according to political orientation. In addition, the increase in the level of measurement invariance at the subsequent steps did not present a significant deterioration of the models’ fit. However, error invariance across political orientations was not achieved because the difference in the CFI between the scalar and error invariance was 0.024 (above the reference values), providing evidence that the CCAS operates similarly across different political orientations (left, center, and right) regarding configural, metric, and scalar invariance (Table 5).

(c)Reliability indices for the CCAS’ factors are displayed in Table 6. No differences between Cronbach’s alpha (α) and McDonald’s omega (ω) were observed. In spite of the fact that the AVE values for cognitive emotional impairment and behavioral engagement are below the reference values, the CCAS is a reliable measure. Also, behavioral engagement presented higher mean values than the other subscales.

(d)Statistically significant differences were found in relation to gender with regard to the climate change anxiety-related cognitive emotional impairment (*t*(533) = −2.079; *p* = 0.007; *d* = 0.555; men, *M* = 1.53, *SD* = 0.54; women, *M* = 1.68, *SD* = 0.56), behavior engagement (*t*(533) = −4.009; *p* < 0.001; *d* = 0.587; men, *M* = 3.53, *SD* = 0.64; women, *M* = 3.77, *SD* = 0.57) and functional impairment (*t*(533) = −2.062; *p* = 0.040; *d* = 0.616; men, *M* = 1.43, *SD* = 0.59; women, *M* = 1.56, *SD* = 0.62) subscales; women presented higher values than men. There are statistically significant differences in the values of the CCAS for personal experience with climate change concerning political orientation (*F*(2, 532) = 3.718; *p* = 0.025; η^2^ = 0.014): left political orientation, *M* = 1.11, *SD* = 0.43; versus center political orientation, *M* = 0.99, *SD* = 0.39; versus right political orientation, *M* = 1.10, *SD* = 0.39), and the post hoc Tukey test showed that these statistically significant differences occurred between the left and the center.

#### 3.3.3. Climate Change Perception Scale

(a)With the aim of validating the climate change perception scale for this population, a confirmatory factor analysis of the original model proposed by the authors was carried out. A second-order model with three factors and eight items presented a very good fit (χ^2^(17) = 1.81; IFI = 0.995; TLI = 0.992; CFI = 0.995; GFI = 0.986; SRMR = 0.015; RMSEA = 0.039 (CI90% LO90 = 0.015; HI90 = 0.061); AIC = 68.78), confirming the authors’ model.(b)Results from the measurement invariance of the CCPS across political orientations are displayed in Table 7. Configural invariance according to political orientation was confirmed during the first step of the multi-group CFAs. However, metric, scalar, and error invariance across political orientations was not achieved because the differences in the RMSEA, CFI, and SRMR values between them were mostly above the reference values, providing evidence that the CCAS operates differently across different political orientations (left, center, and right) (Table 7).

(c)Reliability indices for the CCPS’ factors are displayed in Table 8. No differences between the Cronbach’s alpha (α) and McDonald’s omega (ω) were observed, except for the reality group, whose McDonald’s omega could not be calculated because this group only contains two items. Furthermore, the composite reliability, average variance extracted (AVE), square root of AVE, mean, and standard deviation were calculated (Table 8) and almost all of these values were within the reference range.

(d)Statistically significant differences were found in relation to gender with regard to the CCPS total (*t*(177, 977) = −3.177; *p* = 0.002; *d* = 0.687; men, *M* = 6.30, *SD* = 0.87; women, *M* = 6.59, *SD* = 0.62), reality (*t*(170, 909) = −3.518; *p* < 0.001; *d* = 0.809; men, *M* = 6.36, *SD* = 1.07; women, *M* = 6.71, *SD* = 0.70), causes (*t*(533) = −2.707; *p* = 0.007; *d* = 0.979; men, *M* = 6.13, *SD* = 1.06; women, *M* = 6.40, *SD* = 0.95), and consequences (*t*(192, 759) = −2.396; *p* = 0.018; *d* = 0.718; men, *M* = 6.50, *SD* = 0.83; women, *M* = 6.69, *SD* = 0.68) subscales; women presented higher values than men. Age correlates negatively and significantly with total (*r* = −0.119; *p* = 0.006), reality (*r* = −0.110; *p* < 0.011) and causes (*r* = −0.134; *p* = 0.002) subscales. There are statistically significant differences in the values of the CCPS total (*F*(2, 532) = 4.325; *p* = 0.014; η^2^ = 0.016); CCPS reality (*F*(2, 532) = 4.018; *p* = 0.019; η^2^ = 0.015); and CCPS consequences (*F*(2, 532) = 3.109; *p* = 0.045; η^2^ = 0.012) concerning political orientation. Regarding the total factor, the results for the left political orientation (*M* = 6.65; *SD* = 0.58) versus center political orientation (*M* = 6.51; *SD* = 0.72) versus right political orientation (*M* = 6.43; *SD* = 0.75) from the post hoc Tukey test showed that statistically significant differences occurred between the left and the right. Concerning the reality factor, the results for the left political orientation (*M* = 6.74; *SD* = 0.66) versus center political orientation (*M* = 6.66; *SD* = 0.78) versus right political orientation (*M* = 6.50; *SD* = 0.96) from the post hoc Tukey test showed that the statistically significant differences occurred between the left and the right. Concerning the consequences factor, the results for the left political orientation (*M* = 6.76; *SD* = 0.58) versus center political orientation (*M* = 6.62; *SD* = 0.79) versus right political orientation (*M* = 6.57; *SD* = 0.76) from the post hoc Tukey test showed that the statistically significant differences occurred between the left and the right.

#### 3.3.4. Climate Change Hope Scale

(a)With the goal of validating the author’s model of the climate change hope scale for our sample, a confirmatory factor analysis was carried out and a good model fit was achieved. This model is a unidimensional one, with eight items and three correlations between errors (Table 9).

(b)Results from the measurement invariance of the CCHS across different political orientations are displayed in Table 10. Configural invariance according to political orientation was confirmed during the first step of the multi-group CFA. The small changes in the fit indices at the next steps also supported metric invariance according to political orientation. Additionally, the increase in the level of measurement invariance in the subsequent steps did not present a significant deterioration of the models’ fit; also, error invariance across political orientation was achieved, providing strong evidence that the CCHS operates similarly across different political orientations (left, center, and right). Most of the differences were below 0.01, supporting different levels of measurement equivalence between political orientations (Table 10).

(c)This scale presents a mean value of 4.42 (*SD* = 1.24), Cronbach’s alpha of 0.87, McDonald’s omega of 0.87, composite reliability of 0.90, average variance extracted (AVE) of 0.52, and square root of AVE of 0.72.(d)No differences were found in this scale concerning sociodemographic and political variables.

#### 3.3.5. Climate Change Despair Scale

(a)To validate the authors’ model of the climate change despair scale for our sample, a confirmatory factor analysis was carried out and a good model fit was achieved (χ^2^(2) = 2.26; IFI = 0.996; TLI = 0.987; CFI = 0.996; GFI = 0.996; SRMR = 0.022; RMSEA = 0.049 (CI90% LO90 = 0.000; HI90 = 0.100); AIC = 20.52), confirming the authors’ model.(b)Results from the measurement invariance of the CCDS across different political orientations are displayed in Table 11. Configural invariance according to political orientation was confirmed during the first step of the multi-group CFA. However, metric, scalar, and error invariance across political orientations was not achieved because the differences in the RMSEA, CFI, and SRMR between them were above the reference values, providing little evidence that the CCDS operates differently across different political orientations (left, center, and right) (Table 11).

(c)This model is a unidimensional model, with four items and one correlation between errors. This scale presents a mean of 3.66 (*SD* = 1.42), Cronbach’s alpha of 0.77, McDonald’s omega of 0.78, composite reliability of 0.85, average variance extracted (AVE) of 0.59, and square root of AVE of 0.77.(d)No differences were found in this scale concerning the sociodemographic and political variables.

### 3.4. Associations

The aim of this point is to understand which variables explain each of the pro-environmental behavior subscales.

#### 3.4.1. Correlations

All of the variables under study correlate with each other in a positive and significant way, except the climate change despair scale, which only correlates significantly (albeit with weak correlations) with CCAS cognitive emotional impairment, CCAS functional impairment, CCPS total, and CCPS consequences. Also, CCAS functional impairment only correlates significantly, besides CCDS, with CCPS causes and CCHS total. Lastly, PEBS transportation choice does not correlate with any dimension of the CCPS (Appendix B Table A2).

#### 3.4.2. Regressions

(e)It is mainly age and anxiety behavior engagement that explain, altogether, 36.4% of the outcome’s variable (household behavior) (Table 12). Age, gender, children, anxiety behavior engagement, personal experience with climate change, functional impairment, and hope explain, altogether, 39.4% of the outcome’s variable (information seeking) (Table 12). Lastly, marital status, children, anxiety behavior engagement, and functional impairment explain, altogether, 7.6% of the outcome’s variable (transportation choice) (Table 12).

### 3.5. Moderations

To evaluate the moderating role of climate change perception, despair, and hope in the relationship between climate change anxiety and pro-environmental behavior (f), statistic moderations were performed.

(f)To investigate if despair moderates the relationship between climate change anxiety (cognitive emotional impairment) and pro-environmental behavior (household behavior), a moderator analysis was performed using PROCESS. The outcome variable for this analysis was pro-environmental behavior (household behavior); the predictor variable was climate change anxiety (cognitive emotional impairment); and the moderator variable was despair. The interaction between climate change anxiety (cognitive emotional impairment) and despair was found to be statistically significant (β = −0.07; 95% C.I. (−0.13, −0.01); *p* < 0.05). The conditional effect of climate change anxiety (cognitive emotional impairment) on pro-environmental behavior (household behavior) showed corresponding results. At the low moderation (2.25), the conditional effect was 0.38, with 95% C.I. (0.24, 0.53), and *p* < 0.001; at the medium moderation (3.50), the conditional effect was 0.29, with 95% C.I. (0.19, 0.40), and *p* < 0.001; and at a high moderation (5.25), the conditional effect was 0.17, with 95% C.I. (0.04, 0.30), and *p* < 0.01. These results identify despair as a negative moderator of the relationship between climate change anxiety (cognitive emotional impairment) and pro-environmental behavior (household behavior). The Johnson–Neyman region of significance is 5.63 (below 89.91% and above 10.09%) (Figure 1).

To investigate if despair moderates the relationship between climate change anxiety (personal experience with climate change) and pro-environmental behavior (household behavior), a moderator analysis was performed using PROCESS. The outcome variable for this analysis was pro-environmental behavior (household behavior); the predictor variable was climate change anxiety (personal experience with climate change); and the moderator variable was despair. The interaction between climate change anxiety (personal experience with climate change) and despair was found to be statistically significant (β = −0.11; 95% C.I. (−0.21, −0.02); *p* < 0.05). The conditional effect of climate change anxiety (personal experience with climate change) on pro-environmental behavior (household behavior) showed corresponding results. At a low moderation (2.25), the conditional effect was 0.59, with 95% C.I. (0.40, 0.79), and *p* < 0.001; at medium moderation (3.50), the conditional effect was 0.45, with 95% C.I. (0.32, 0.59), and *p* < 0.001; and at a high moderation (5.25), the conditional effect was 0.17, with 95% C.I. (0.04, 0.30), and *p* < 0.05. These results identify despair as a negative moderator of the relationship between climate change anxiety (personal experience with climate change) and pro-environmental behavior (household behavior). The Johnson–Neyman region of significance is 5.57 (below 89.91% and above 10.09%) (Figure 2).

To investigate if hope moderates the relationship between climate change anxiety (behavior engagement) and pro-environmental behavior (information seeking), a moderator analysis was performed using PROCESS. The outcome variable for this analysis was pro-environmental behavior (information seeking); the predictor variable was climate change anxiety (behavior engagement); and the moderator variable was hope. The interaction between climate change anxiety (behavior engagement) and hope was found to be statistically significant (β = 0.08; 95% C.I. (0.00, 0.17); *p* < 0.05). The conditional effect of climate change anxiety (behavior engagement) on pro-environmental behavior (information seeking) showed corresponding results. At a low moderation (3.13), the conditional effect was 0.63, with 95% C.I. (0.46, 0.80), and *p* < 0.001; at a medium moderation (4.50), the conditional effect was 0.75, with 95% C.I. (0.61, 0.89), and *p* < 0.001; at a high moderation (5.75), the conditional effect was 0.85, with 95% C.I. (0.67, 1.04), and *p* < 0.001. These results identify hope as a positive moderator of the relationship between climate change anxiety (behavior engagement) and pro-environmental behavior (information seeking). The Johnson–Neyman region of significance is 0.02 (below 0.19% and above 99.81%) (Figure 3).

To investigate if climate change perception (consequences) moderates the relationship between climate change anxiety (behavior engagement) and pro-environmental behavior (information seeking), a moderator analysis was performed using PROCESS. The outcome variable for this analysis was pro-environmental behavior (information seeking); the predictor variable was climate change anxiety (behavior engagement); and the moderator variable was climate change perception (consequences). The interaction between climate change anxiety (behavior engagement) and climate change perception (consequences) was found to be statistically significant (β = 0.21; 95% C.I. (0.08, 0.33); *p* < 0.05). The conditional effect of climate change anxiety (behavior engagement) on pro-environmental behavior (information seeking) showed corresponding results. At a low moderation (6.00), the conditional effect was 0.71, with 95% C.I. (0.56, 0.81), and *p* < 0.001; and at medium and high moderations (7.00), the conditional effects were 0.91, with 95% C.I. (0.77, 1.06), and *p* < 0.001. These results identify climate change perception (consequences) as a positive moderator of the relationship between climate change anxiety (behavior engagement) and pro-environmental behavior (information seeking). The Johnson–Neyman region of significance is 4.17 (below 2.24% and above 97.76%) (Figure 4).

## 4. Discussion

The main objective of this paper was to assess pro-environmental behavior and climate change perception, anxiety, hope, and despair across different political orientations. Our specific aims included validating all of the instrumentation; assessing whether the factor structure of the scales were valid across different political orientations; evaluating their reliability; assessing differences concerning age, gender, and political orientation; learning the variables that explain each of the pro-environmental behavior subscales; and evaluating the moderating role of climate change perception, despair, and hope in the relationship between climate change anxiety and pro-environmental behavior.

All validated instruments showed a good fit (confirming hypothesis one) and good reliability, convergence, and discriminant validity indicators (confirming hypothesis three). Regarding the pro-environmental behavior scale (PEBS), it was verified that the Cronbach’s alpha value for the total of our study is higher than that of the original version [64], and the Cronbach’s alpha values of the subscales are also higher than that of the original version, with the exception of the transportation choice subscale. Concerning the climate change anxiety scale (CCAS), we found that the Cronbach’s alpha value of our study is higher than that of the original version [17]. This scale, as in the original version, consists of twenty-two items and four subscales (cognitive emotional impairment, behavioral engagement, personal experience with climate change, and functional impairment). The average, taking into account all subscales, is higher in Factor 2 and lower in Factor 4. Regarding the climate change hope scale (CCHS), we found that the Cronbach’s alpha value in our study is higher than that of the original version [64]. This instrument, as in the original version, is composed of eight items. With regard to the climate change perception scale (CCPS), it was also verified that the Cronbach’s alpha value for the different subscales in our study is higher than that of the original version [35], with the exception of the reality subscale. The same applies to the climate change despair scale (CCDS), with the total Cronbach’s alpha value of our study being higher than that of the original version [64].

Pro-environmental behavior and climate change hope achieved the assessed four levels of invariance across different political orientations; climate change anxiety achieved the first three levels of invariance; climate change perception and climate change despair achieved only configural invariance (partially confirming hypothesis two). Also, Wang et al. [88] developed the psychological ownership of nature (PON) concept and found measurement invariance across gender, age, and political subgroups.

Moreover, climate change anxiety, personal experience, and climate change perception total, reality and consequences present higher values in the left political orientation than in the right or the center (confirming hypothesis four). Poortinga and collaborators [33] found a negative association between right-wing political orientations and concerns and anxiety about climate change. Previous studies have shown that political orientation can moderate the relationship between perceptions of climate change and anxiety derived from it [89,90]. For example, Hamilton [89] found that anxiety surrounding the impacts of climate change increased when education levels were higher for left-wing individuals, while it decreased for those who identified as right-wing individuals. Malka et al. [91] concluded that a greater knowledge about climate change was related to greater concern and anxiety among Democrats, which was not the case for Republicans. These studies indicate that individuals can filter information in a way that is aligned with their political ideology [90].

Regarding the perception of climate change and according to the System Justification Theory [92], individuals who have a right-wing political ideology tend to justify the existing social and economic system, while left-wing individuals tend to question and challenge that same system. In this way, right-wing individuals may be more likely to deny climate change since these changes may imply the need for changes in the existing economic and social system. On the other hand, left-wing individuals may be more likely to accept the reality of climate change and believe in its consequences, as they believe that changes are necessary to achieve a fairer and more equitable world [53,56]. Exposure to information sources can also influence this relationship, with a study by Schäfer and Painter [93] stating that individuals who watch television more tend to be more skeptical about climate change, while those who read more newspapers tend to be less skeptical. This can be explained by the fact that television often presents the controversy regarding climate change with a balanced debate between supporters and skeptics, while newspapers tend to present scientific results in a more objective and factual way [93].

Climate change anxiety variables contribute the most to explaining pro-environmental behaviors (confirming hypothesis five). Results found in other studies [59,60,61] have also concluded that anxiety about climate change may be linked to pro-environmental behaviors. For example, when individuals are more alert to danger, they are more motivated to take action, that is, climate anxiety can serve as a source of motivation to stimulate their environmental commitment [61].

Moreover, hope (positively), despair (negatively), and climate change perception consequences (positively) moderate the relationship between climate change anxiety and pro-environmental behavior. This means that the less despair people feel about climate change, the stronger the relationship between climate change anxiety and pro-environmental behaviors. On the contrary, the more hope and climate change perception people have, the stronger the relationship between climate change anxiety and pro-environmental behaviors (confirming hypothesis six). These results are in line with those of Stevenson and Peterson [64], who consider that both concepts (despair and hope) have a future-oriented nature; one increasing and the other decreasing pro-environmental behaviors. Also, McKinnon [65] and Nairn [66] consider that while hope motivates individuals to join sustainability-promoting groups, despair can be a demotivating factor [65,66]. Most individuals have limited knowledge about how regimes, organizations, laws, international environmental treaties, and negotiations proceed and are therefore less likely to make judgments regarding their mitigation and adaptation objectives; the most common form of despair in the face of climate change is the feeling that it is impossible to make a difference by reducing greenhouse gas emissions at the individual level, often followed by the thought that it is better not to try [65].

According to the study by Pickering and Dale [94], individuals with higher levels of anxiety are more concerned with the world and, therefore, adopt more pro-environmental views. Indeed, studies by Clayton [26] and Clayton and Karazsia [17] demonstrated a strong link between general anxiety levels and climate anxiety. In another recent large-scale study covering 28 countries, Ogunbode and collaborators [95] concluded that climate anxiety was positively associated with increased pro-environmental attitudes and that, in richer democratic countries, such anxiety was positively associated with engagement in pro-environmental behaviors.

Statistically significant differences were found in relation to gender with regard to pro-environmental behavior (household behavior and information-seeking behavior); climate change anxiety (cognitive emotional impairment, behavior engagement, and functional impairment); and climate change perception (total and all subscales): women presented higher values than men in all dimensions (confirming hypothesis four). Psychology may offer an explanation for this gender difference in environmental behavior. Studies suggest that gender identity is an important factor in determining environmental behaviors [15,72,73]. Women tend to be more likely to exhibit pro-environmental behaviors because sustainability and concern for the environment are consistent with values traditionally associated with femininity, such as care and social responsibility [74,75], indicating that women are more exposed to the social norms and cultural expectations associated with caring for and preserving the environment, which can lead to a greater adoption of pro-environmental behaviors [15,76]. This gender difference has already been reported in several studies. A meta-analysis performed by Li et al. [15] found that women were more likely to exhibit pro-environmental behaviors compared to men. Other studies also point in the same direction, such as those by Smith et al. [74] and Xiao and McCright [75].

Age correlates positively and significantly with pro-environmental behavior (household behavior and information-seeking behavior), and climate change perception (total, reality, and causes) (confirming hypothesis four): older people tend to present more pro-environmental behavior than younger people, and a higher climate change perception (total, reality, and causes). Several factors can explain this trend, and scientific evidence points to some possibilities. One of them is that older people have greater environmental awareness and social responsibility towards the environment. This can be explained by Kohlberg’s Theory of Moral Development, which suggests that the highest stage of moral development is that of social responsibility, in which people recognize the importance of contributing to the well-being of society and the environment [77]. Therefore, it is possible that age is related to a person’s level of moral development and, consequently, to their environmental concern [77]. Another factor that may explain this trend towards an increase in pro-environmental behavior at older ages is social and cultural influence. Modern society has become increasingly concerned with sustainability and the preservation of the environment, and this concern may be being transmitted from generation to generation [78]. In addition, the media and environmental education also play a crucial role in shaping environmental awareness and promoting sustainable behavior [78].

The results of this study have theoretical and managerial implications. Theoretical frameworks should integrate cognitive, emotional, and motivational aspects to capture the complexity of individuals’ responses to climate change [96]. These frameworks should account for the reciprocal influence between behavioral choices and psychological states, assessing changes in perceptions over time and understanding how they influence behavioral outcomes [97]. Theoretical models should incorporate emotional issues to better explain the motivations behind pro-environmental actions [97]. Lastly, theoretical models should consider the influence of contextual factors, such as socioeconomic status, cultural background, and geographic location, on climate change perception and pro-environmental behaviors [96]. Concerning managerial implications, organizations and policymakers can use insights from research in this field to develop tailored communication strategies [98]. Efforts to counter climate change despair can involve positive narratives about successful environmental initiatives, highlighting the impact of pro-environmental behaviors, and showcasing examples of sustainable living [98]. Managerial strategies should prioritize education and awareness campaigns that enhance the public’s understanding of climate change.

This study has some limitations that must be addressed, Our sample is not representative of the Portuguese population; in fact, the sampling method is a limitation of this study, and this issue could introduce bias and undermine the robustness of the results. Thus, replications of this validation study with a sample that meets these requirements may be considered in future studies. In addition, the investigative protocol was applied online, which means that it was not possible to clarify any doubts that might arise. Therefore, future studies may also be conducted in person.

## 5. Conclusions

The main objective of this paper was to assess pro-environmental behavior and climate change perception, anxiety, hope, and despair across different political orientations. Confirmatory factor analyses (CFAs), multi-group CFAs, multiple linear regressions, and moderations were performed. The results showed that pro-environmental behavior and climate change hope achieved the four assessed levels of invariance across different political orientations; climate change anxiety achieved the first three levels of invariance; and climate change perception and climate change despair achieved only configural invariance. Also, climate change anxiety, personal experience with climate change, and climate change perception (total, reality, and consequences) presented higher values in the left political orientation that in the right or the center. Climate change anxiety variables contributed most to explaining individuals’ pro-environmental behaviors. Furthermore, hope, despair, and climate change perception (consequences) moderated the relationship between climate change anxiety and pro-environmental behaviors. These results add new data to the literature on this subject and open up new avenues for investigation, namely for understanding why high levels of anxiety lead to more pro-environmental behaviors.

## Figures and Tables

**Figure 1 behavsci-13-00966-f001:**
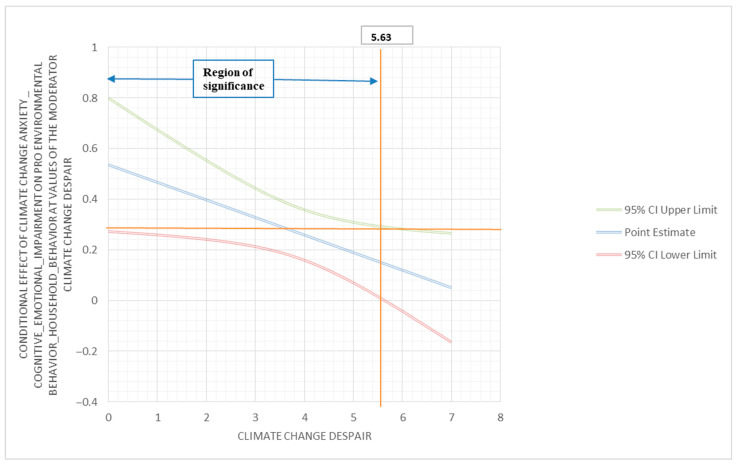
Conditional effect of climate change anxiety (cognitive emotional impairment) on pro-environmental behavior (household behavior) at values of the climate change despair moderator.

**Figure 2 behavsci-13-00966-f002:**
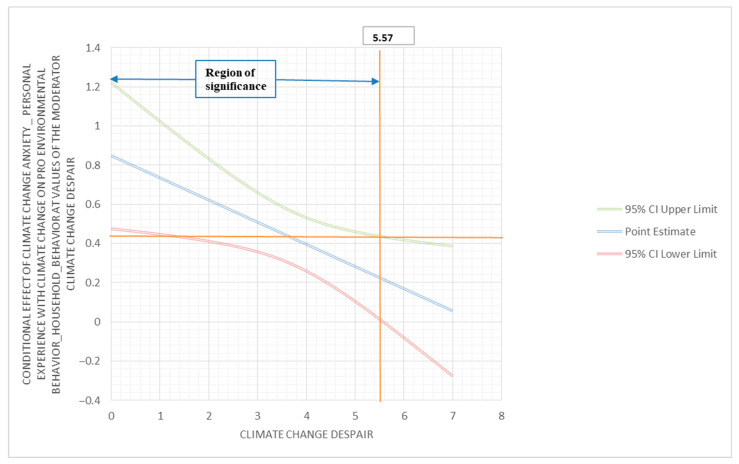
Conditional effect of climate change anxiety (personal experience with climate change) on pro-environmental behavior (household behavior) at values of the climate change despair moderator.

**Figure 3 behavsci-13-00966-f003:**
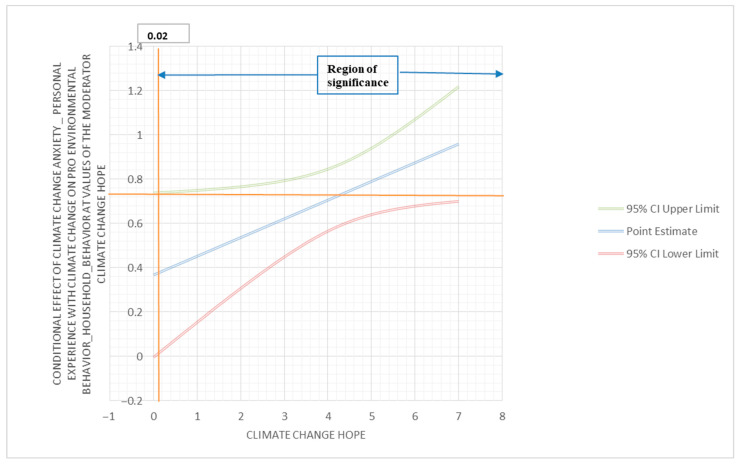
Conditional effect of climate change anxiety (personal experience with climate change) on pro-environmental behavior (household behavior) at values of the climate change hope moderator.

**Figure 4 behavsci-13-00966-f004:**
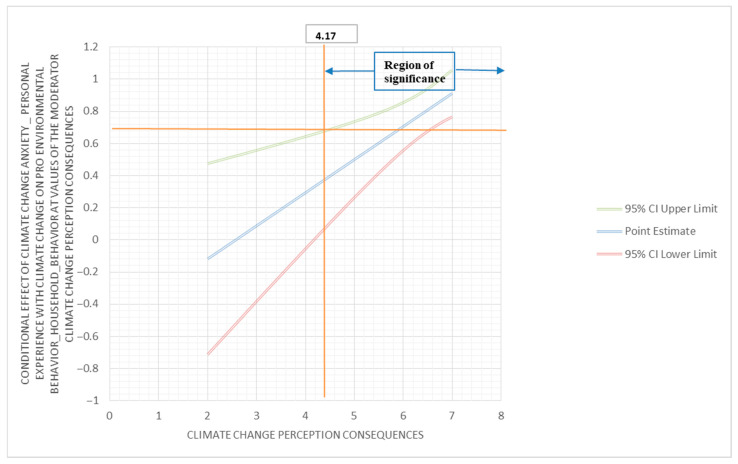
Conditional effect of climate change anxiety (personal experience with climate change) on pro-environmental behavior (household behavior) at values of the climate change perception (consequences) moderator.

**Table 1 behavsci-13-00966-t001:** Goodness of fit indices for the four-dimensional model of the pro-environmental behavior scale (*N* = 535).

									RMSEA CI 90%			
	χ^2^	DF	χ^2^/DF	IFI	TLI	CFI	GFI	SRMS	RMSEA	LO90	HI90	AIC
Three factors, 10 items	226.88	32	7.09	0.878	0.828	0.877	0.913	0.079	0.107	0.094	0.120	272.88
Three factors, 10 items, three correlations between errors	79.99	29	2.76	0.968	0.950	0.968	0.971	0.050	0.057	0.043	0.063	131.99

Note: χ^2^ = qui-squared; DF = degrees of freedom; IFI = incremental fit index; TLI = Tucker–Lewis index; CFI = comparative fit index; GFI = goodness of fit index; SRMS = standard root mean square; RMSEA = root mean square error of approximation; CI = confidence interval; AIC = Akaike information criterion.

**Table 2 behavsci-13-00966-t002:** Multi-group CFAs of the pro-environmental behavior scale according to political orientation (*N* = 535).

	χ^2^	*DF*	χ^2^/*DF*	RMSEA (CI)	CFI	IFI	SRMS	Comparisons	Δ RMSEA	Δ CFI	Δ SRMR
Configural invariance	132.56	87	1.52	0.031 (0.020–0.042)	0.971	0.972	0.062	NA	NA	NA	NA
Metric invariance	156.47	101	1.54	0.032 (0.022–0.042)	0.965	0.966	0.063	Configural vs. metric	0.001	0.006	0.001
Scalar invariance	167.87	113	1.49	0.030 (0.020–0.039)	0.966	0.966	0.064	Metric vs. scalar	0.002	0.001	0.001
Error variance invariance	198.94	139	1.43	0.028 (0.019–0.037)	0.962	0.962	0.065	Scalar vs. error variance	0.002	0.004	0.001

Note: χ^2^ = qui-squared; DF = degrees of freedom; IFI = incremental fit index; CFI = comparative fit index; RMSEA = root mean square error of approximation; CI = confidence interval; SRMS = standard root mean square; ΔRMSEA = change in the RMSEA compared with the previous model (expressed in absolute values); ΔCFI = change in the CFI compared with the previous model (expressed in absolute values); ΔSRMR = change in the SRMR compared with the previous model (expressed in absolute values). All models are significant at *p* < 0.001. NA = not applicable.

**Table 3 behavsci-13-00966-t003:** Pearson’s correlations Cronbach’s alpha, McDonald’s omega, composite reliability, average variance extracted (AVE), AVE square roots, mean, and standard deviation for the pro-environmental behavior scale (*N* = 535).

	Pearson’s Correlations							
	1	2	3	α	ω	CR	AVE	Mean (*SD*)
1. Household behavior	**0.67**			0.68	0.64	0.80	0.45	4.18 (0.66)
2. Information-seeking behavior	0.428 **	**0.87**		0.84	0.85	0.90	0.76	2.69 (1.01)
3. Transportation choice	0.276 **	0.284 **	**0.81**	0.48	^a^	0.79	0.66	2.57 (0.92)

Note: ** = *p* < 0.001; α = Cronbach’s alpha; ω = McDonald’s omega; CR = composite reliability; AVE = average variance extracted; **bold** (diagonal) = AVE square roots; *SD* = standard deviation; ^a^ = McDonald’s omega cannot be calculated because the number of items is less than three.

**Table 4 behavsci-13-00966-t004:** Goodness of fit indices for the four-dimensional model of the climate change anxiety scale (*N* = 535).

										RMSEA CI 90%		
	χ^2^	DF	χ^2^/DF	IFI	TLI	CFI	GFI	SRMS	RMSEA	LO90	HI90	AIC
Second-order model, four factors, 22 items	742.24	205	3.62	0.873	0.856	0.872	0.879	0.078	0.070	0.065	0.076	838.24
Second-order model, four factors, 22 items, six correlations between errors	485.66	199	2.44	0.932	0.921	0.932	0.922	0.061	0.052	0.046	0.058	593.66
Four factors, 22 items	715.39	203	3.52	0.879	0.861	0.878	0.881	0.074	0.069	0.063	0.074	815.39
Four factors, 22 items, six correlations between errors	467.31	197	2.37	0.936	0.924	0.936	0.925	0.057	0.051	0.045	0.057	579.31

Note: χ^2^ = qui-squared; DF = degrees of freedom; IFI = incremental fit index; TLI = Tucker–Lewis index; CFI = comparative fit index; GFI = goodness of fit index; SRMS = standard root mean square; RMSEA = root mean square error of approximation; CI = confidence interval; AIC = Akaike information criterion.

**Table 5 behavsci-13-00966-t005:** Multi-group CFA of the climate change anxiety scale according to political orientation (*N* = 535).

	χ^2^	*DF*	χ^2^/*DF*	RMSEA (CI)	CFI	IFI	SRMS	Comparisons	Δ RMSEA	Δ CFI	Δ SRMR
Configural invariance	963.13	591	1.63	0.034 (0.030–0.038)	0.914	0.916	0.084	NA	NA	NA	NA
Metric invariance	1008.80	627	1.61	0.034 (0.030–0.038)	0.912	0.913	0.091	Configural vs. metric	0.000	0.002	0.007
Scalar invariance	1035.26	647	1.60	0.034 (0.030–0.037)	0.911	0.911	0.095	Metric vs. scalar	0.000	0.001	0.004
Error variance invariance	1195.64	703	1.70	0.036 (0.033–0.040)	0.887	0.886	0.093	Scalar vs. error variance	0.002	0.024	0.002

Note: χ^2^ = qui-squared; DF = degrees of freedom; IFI = incremental fit index; CFI = comparative fit index; RMSEA = root mean square error of approximation; CI = confidence interval; SRMS = standard root mean square; ΔRMSEA = change in the RMSEA compared with the previous model (expressed in absolute values); ΔCFI = change in the CFI compared with the previous model (expressed in absolute values); ΔSRMR = change in the SRMR compared with the previous model (expressed in absolute values). All models are significant at *p* < 0.001. NA = not applicable.

**Table 6 behavsci-13-00966-t006:** Pearson’s correlations Cronbach’s alpha, McDonald’s omega, composite reliability, average variance extracted (AVE), AVE square roots, mean, and standard deviation for the climate change anxiety scale (*N* = 535).

	Pearson’s Correlations								
	1	2	3	4	α	ω	CR	AVE	Mean (*SD*)
1. Cognitive emotional impairment	**0.68**				0.83	0.83	0.87	0.46	1.64 (0.56)
2. Behavioral engagement	0.324 **	**0.63**			0.66	0.65	0.79	0.40	3.71 (0.59)
3. Personal experience with climate change	0.481 **	0.408 **	**0.83**		0.77	0.78	0.87	0.69	1.05 (0.41)
4. Functional impairment	0.651 **	0.262 **	0.487 **	**0.80**	0.85	0.85	0.90	0.64	1.53 (0.62)

Note: ** = *p* < 0.001; α = Cronbach’s alpha; ω = McDonald’s omega; CR = composite reliability; AVE = average variance extracted; **bold** (diagonal) = AVE square roots; *SD* = standard deviation.

**Table 7 behavsci-13-00966-t007:** Multi-group CFA of the climate change perception scale according to political orientation (*N* = 535).

	χ^2^	*DF*	χ^2^/*DF*	RMSEA (CI)	CFI	IFI	SRMS	Comparisons	Δ RMSEA	Δ CFI	Δ SRMR
Configural invariance	112.62	51	2.21	0.048 (0.036–0.060)	0.980	0.980	0.060	NA	NA	NA	NA
Metric invariance	159.88	61	2.62	0.055 (0.045–0.066)	0.967	0.968	0.105	Configural vs. metric	0.007	0.013	0.045
Scalar invariance	186.08	65	2.86	0.059 (0.049–0.069)	0.960	0.960	0.104	Metric vs. scalar	0.004	0.007	0.001
Error variance invariance	200.55	67	2.99	0.061 (0.052–0.071)	0.956	0.956	0.085	Scalar vs. error variance	0.003	0.004	0.019

Note: χ^2^ = qui-squared; DF = degrees of freedom; IFI = incremental fit index; CFI = comparative fit index; RMSEA = root mean square error of approximation; CI = confidence interval; SRMS = standard root mean square; ΔRMSEA = change in the RMSEA compared with the previous model (expressed in absolute values); ΔCFI = change in the CFI compared with the previous model (expressed in absolute values); ΔSRMR = change in the SRMR compared with the previous model (expressed in absolute values). All models are significant at *p* < 0.001. NA = not applicable.

**Table 8 behavsci-13-00966-t008:** Pearson’s correlations Cronbach’s alpha, McDonald’s omega, composite reliability, average variance extracted (AVE), AVE square roots, mean, and standard deviation for the climate change perception scale (*N* = 535).

	Pearson’s Correlations								
	1	2	3	4	α	ω	CR	AVE	Mean (*SD*)
1. Total	**0.75**				0.88	0.88	0.91	0.57	6.52 (0.70)
2. Reality	0.688 **	**0.86**			0.62	^a^	0.85	0.74	6.63 (0.82)
3. Causes	0.873 **	0.369 **	**0.94**		0.94	0.94	0.96	0.89	6.33 (0.99)
4. Consequences	0.856 **	0.504 **	0.598 **	**0.89**	0.87	0.87	0.92	0.80	6.64 (0.72)

Note: ** = *p* < 0.001; α = Cronbach’s alpha; ω = McDonald’s omega; CR = composite reliability; AVE = average variance extracted; **bold** (diagonal) = AVE square roots; *SD* = standard deviation; ^a^ = McDonald’s omega cannot be calculated because the number of items is less than three.

**Table 9 behavsci-13-00966-t009:** Goodness of fit indices for the unidimensional model of the climate change hope scale (*N* = 535).

									RMSEA CI90%			
	χ^2^	DF	χ^2^/DF	IFI	TLI	CFI	GFI	SRMS	RMSEA	LO90	HI90	AIC
One factor, 8 items	166.38	20	8.32	0.911	0.875	0.911	0.922	0.053	0.117	0.101	0.134	198.38
One factor, 8 items, three correlations between errors	53.68	17	3.16	0.978	0.963	0.978	0.975	0.030	0.064	0.045	0.083	91.68

Note: χ^2^ = qui-squared; DF = degrees of freedom; IFI = incremental fit index; TLI = Tucker–Lewis index; CFI = comparative fit index; GFI = goodness of fit index; SRMS = standard root mean square; RMSEA = root mean square error of approximation; CI = confidence interval; AIC = Akaike information criterion.

**Table 10 behavsci-13-00966-t010:** Multi-group CFA of the climate change hope scale according to political orientation (*N* = 535).

	χ^2^	*DF*	χ^2^/*DF*	RMSEA (CI)	CFI	IFI	SRMS	Comparisons	Δ RMSEA	Δ CFI	Δ SRMR
Configural invariance	114.82	51	2.25	0.048 (0.037–0.060)	0.962	0.962	0.041	NA	NA	NA	NA
Metric invariance	121.87	65	1.88	0.041 (0.029–0.052)	0.966	0.966	0.045	Configural vs. metric	0.007	0.004	0.003
Scalar invariance	123.88	67	1.85	0.040 (0.029–0.051)	0.966	0.966	0.045	Metric vs. scalar	0.001	0.000	0.000
Error variance invariance	146.84	89	1.65	0.035 (0.025–0.045)	0.965	0.965	0.054	Scalar vs. error variance	0.005	0.001	0.009

Note: χ^2^ = qui-squared; DF = degrees of freedom; IFI = incremental fit index; CFI = comparative fit index; RMSEA = root mean square error of approximation; CI = confidence interval; SRMS = standard root mean square; ΔRMSEA = change in the RMSEA compared with the previous model (expressed in absolute values); ΔCFI = change in the CFI compared with the previous model (expressed in absolute values); ΔSRMR = change in the SRMR compared with the previous model (expressed in absolute values). All models are significant at *p* < 0.001. NA = not applicable.

**Table 11 behavsci-13-00966-t011:** Multi-group CFA of the climate change despair scale according to political orientation (*N* = 535).

	χ^2^	*DF*	χ^2^/*DF*	RMSEA (CI)	CFI	IFI	SRMS	Comparisons	Δ RMSEA	Δ CFI	Δ SRMR
Configural invariance	8.10	6	1.35	0.026 (0.000–0.066)	0.996	0.996	0.029	NA	NA	NA	NA
Metric invariance	8.86	10	0.89	0.000 (0.000–0.043)	1.000	1.002	0.032	Configural vs. metric	0.026	0.004	0.003
Scalar invariance	11.76	12	0.98	0.000 (0.000–0.043)	1.000	1.000	0.037	Metric vs. scalar	0.000	0.000	0.005
Error variance invariance	28.46	22	1.29	0.023 (0.000–0.043)	0.989	0.989	0.037	Scalar vs. error variance	0.023	0.011	0.000

Note: χ^2^ = qui-squared; DF = degrees of freedom; IFI = incremental fit index; CFI = comparative fit index; RMSEA = root mean square error of approximation; CI = confidence interval; SRMS = standard root mean square; ΔRMSEA = change in the RMSEA compared with the previous model (expressed in absolute values); ΔCFI = change in the CFI compared with the previous model (expressed in absolute values); ΔSRMR = change in the SRMR compared with the previous model (expressed in absolute values). All models are significant at *p* < 0.001. NA = not applicable.

**Table 12 behavsci-13-00966-t012:** Variables that contribute to explaining the results of the pro-environmental behavior scale.

	Household Behavior			Information-Seeking Behavior			Transportation Choice		
	B	EP B	β	B	EP B	β	B	EP B	β
Age	0.004	0.002	0.072	0.009	0.005	0.097			
Gender				0.253	0.084	0.108			
Marital status							−0.240	0.121	−0.123
Children				0.291	0.135	0.102	0.360	0.161	0.138
CCAS behavioral engagement	0.669	0.039	0.598	0.480	0.068	0.281	0.284	0.067	0.183
CCAS personal experience with climate change				0.347	0.103	0.139			
CCAS functional impairment				0.493	0.064	0.301	0.239	0.064	0.160
CCHS total				0.071	0.030	0.088			
R^2^ (R^2^ Adj.)	0.366 (**0.364**)			0.403 (**0.395**)			0.082 (**0.076**)		
F for change in R^2^	300.339 **			75.167 **			21.066 **		

Note: R^2^ = R squared; R^2^ Adj. = R squared adjusted; B = unstandardized regression coefficients; EP B = unstandardized error of B; β = standardized regression coefficients; ** = *p* < 0.001; **bold** = the value of R^2^ adjusted.

## Data Availability

The data presented in this study are available on request from the corresponding author. The data are not publicly available due to privacy issues.

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
