# Peer review of "Pro-Environmental Behavior and Climate Change Anxiety, Perception, Hope, and Despair According to Political Orientation"

_behavsci, 2023, doi:10.3390/bs13120966_

Round 1
Reviewer 1 Report
Comments and Suggestions for Authors
The manuscript I reviewed in the journal is of high quality. The authors conducted a study that validated several scales related to Personal Ecological Behavior (PEB), eco-anxiety, climate change perception, climate hope, and despair. They also explored the possible relationships between these variables and specific demographic factors. Overall, I recommend some revisions. Here are my specific comments:
Line 101 - It would be beneficial if the authors could provide citations for the assertion that climate change can be perceived as "positive".
The authors' literature review was thorough, particularly in regard to environmental behavior, eco-anxiety, and climate change perceptions. However, I suggest extending the discussion on the relationship between hope/despair and climate change and environmental behaviors in section 1.8. As Stevens and Peterson (2016) argued, both concepts have a future-oriented aspect that could be explored further. Perhaps the authors could also discuss generativity/generative concern in the context of environmental behavior, where individuals think about future generations or aim to leave a positive impact for the future.
I recommend moving section 2.1 to the end of the introduction for better flow and coherence.
Furthermore, I encourage the authors to expand on certain sections. It would be helpful to provide the rationales, research gaps, specific research questions, and hypotheses more explicitly. Additionally, the authors should justify why they chose to validate the scales in the Portuguese population.
I am uncertain about aim "f" and its proposal of moderation. I suggest the authors provide a more comprehensive explanation of this aim.
Considering that there are numerous scales related to PEB, I recommend the authors provide justification for selecting the Stevenson and Peterson (2015) scale for validation. Furthermore, it appears that this particular scale has low reliability. Have previous studies in Portugal frequently used it?
Regarding the climate change anxiety scale, I noticed that the 22 items used are not all specifically related to "anxiety". Some items capture behavioral changes and experiences related to climate change, as evidenced by Clayton and Karazsia's (2020) study. It would be helpful if the authors could clarify this issue and make note of the inclusion of these subscales. Additionally, it is worth mentioning that the mean scores for behavioral engagement are higher than those for anxiety and impairment.
The figures in the manuscript are somewhat confusing due to their curved, nonlinear nature. Additionally, the variable labels are difficult to read. It would be beneficial to improve the clarity of the figures and enhance the legibility of the variable labels.
Comments on the Quality of English LanguageMinor editing.
Reviewer 2 Report
Comments and Suggestions for Authors
Dear authors,
I had the pleasure of revising your very interesting and relevant article. I particularly appreciated your introduction, which effectively introduces the theoretical questions and supports the main motivation behind the study. However, it may be a bit too long and detailed for the purpose, and could benefit from being more concise. Nevertheless, I don't feel inclined to suggest specific cuts, so the decision is ultimately up to you.
I also found your empirical strategy commendable and appreciated the effort you put into describing it in great detail. However, I must point out that section 3.3 is quite confusing and needs to be revised. Firstly, the section title is too long and could easily be mistaken as a sublabel for table 1. I suggest shortening it and including relevant information in the main text instead. Furthermore, section 3.3 contains an excessive number of tables. It might be more appropriate to move the majority of them to an appendix and use the section to describe your approach and findings in a more narrative manner.
Another important issue is that in sections 2.1 and 2.2, you assume the concept of scales without clearly defining them. This becomes clear only in section 2.3. It is crucial to clearly define concepts before utilizing them to avoid confusion.
I am also uncertain about the meaning of "XXX" at line 230. Is it a typo? Does it refer to an ethical committee or an academic journal? If you are unable to clarify this at this stage of the revision, it is acceptable. However, please provide a broad understanding of what is being referred to.
Finally, I believe you should discuss your sampling method as a limitation of the article. Conducting snowball sampling with only 500 observations results in an unbalanced sample that cannot be considered probabilistic. This issue could introduce bias and undermine the robustness of your results.
Apart from these suggestions, I found the article to be well-written and interesting, and I believe it warrants further exploration through additional research.
Reviewer 3 Report
Comments and Suggestions for Authors
Dear Authors,
in my opinion your paper is interesting and well written. I have only some suggestions to improve it.
First of all , you must better highlight the gap that the research intends to fill.
In the materials and methods section, it's important to add the period in which the research was carried out.
The results and their discussions are clear.
In the conclusion you must deepen the theoretical and managerial implication of the research.
Finally, you must add more updated references.
Good luck!
Round 2
Reviewer 1 Report
Comments and Suggestions for Authors
The authors have addressed all my comments.